# Green Synthesis of Titanium Dioxide Nanoparticles Using *Ocimum sanctum* Leaf Extract: In Vitro Characterization and Its Healing Efficacy in Diabetic Wounds

**DOI:** 10.3390/molecules27227712

**Published:** 2022-11-09

**Authors:** Mohammad Zaki Ahmad, Ali S. Alasiri, Javed Ahmad, Abdulsalam A. Alqahtani, Md Margub Abdullah, Basel A. Abdel-Wahab, Kalyani Pathak, Riya Saikia, Aparoop Das, Himangshu Sarma, Seham Abdullah Alzahrani

**Affiliations:** 1Department of Pharmaceutics, College of Pharmacy, Najran University, Najran 11001, Kingdom of Saudi Arabia; 2Advanced Materials and Nano-Research Centre, Department of Physics, Faculty of Science and Arts, Najran University, Najran 11001, Kingdom of Saudi Arabia; 3Department of Pharmacology, College of Pharmacy, Najran University, Najran 11001, Kingdom of Saudi Arabia; 4Department of Pharmacology, College of Medicine, Assiut University, Assiut 7111, Egypt; 5Department of Pharmaceutical Sciences, Dibrugarh University, Dibrugarh 786004, Assam, India; 6Sophisticated Analytical Instrument Facility (SAIF), Girijananda Chowdhury Institute of Pharmaceutical Science (GIPS), Guwahati 781017, Assam, India; 7Pharmacy Department, Khamis Mushait General Hosptial, King Khalid Rd, Al Shifa, Khamis Mushait 62433, Kingdom of Saudi Arabia

**Keywords:** *Ocimum sanctum* leaf extract, titanium dioxide nanoparticles (TiO_2_ NPs), green synthesis, energy dispersive X-ray (EDX) analysis, chitosan gel, diabetic wounds

## Abstract

Diabetes mellitus is one of the most prevalent metabolic disorders characterized by hyperglycemia due to impaired glucose metabolism. Overproduction of free radicals due to chronic hyperglycemia may cause oxidative stress, which delays wound healing in diabetic conditions. For people with diabetes, this impeded wound healing is one of the predominant reasons for mortality and morbidity. The study aimed to develop an *Ocimum sanctum* leaf extract-mediated green synthesis of titanium dioxide (TiO_2_) nanoparticles (NPs) and further incorporate them into 2% chitosan (CS) gel for diabetic wound healing. UV-visible spectrum analysis recorded the sharp peak at 235 and 320 nm, and this was the preliminary sign for the biosynthesis of TiO_2_ NPs. The FTIR analysis was used to perform a qualitative validation of the biosynthesized TiO_2_ nanoparticles. XRD analysis indicated the crystallinity of TiO_2_ NPs in anatase form. Microscopic investigation revealed that TiO_2_ NPs were spherical and polygonal in shape, with sizes ranging from 75 to 123 nm. The EDX analysis of green synthesized NPs showed the presence of TiO_2_ NPs, demonstrating the peak of titanium ion and oxygen. The hydrodynamic diameter and polydispersity index (PDI) of the TiO_2_ NPs were found to be 130.3 nm and 0.237, respectively. The developed TiO_2_ NPs containing CS gel exhibited the desired thixotropic properties with pseudoplastic behavior. In vivo wound healing studies and histopathological investigations of healed wounds demonstrated the excellent wound-healing efficacy of TiO_2_ NPs containing CS gel in diabetic rats.

## 1. Introduction

Diabetes mellitus (DM) is characterized by elevated blood sugar levels (hyperglycemia) that are brought on by dysfunction in the production or action of insulin or both [1]. Chronic hyperglycemia brought on by diabetes is associated with the long-term damage/failure of multiple organs [1]. According to the International Diabetes Federation, 537 million adults have DM, which is expected to rise to 643 million by 2030 and 783 million by 2045 [2].

Diabetes ketoacidosis (DKA) and hyperosmolar hyperglycemic nonketotic coma (HONK) are two conditions that are typically categorized as acute complications of DM [3]. Diabetic neuropathy, diabetic retinopathy, macrovascular disease, and microvascular diseases are considered chronic complications of DM [3,4]. Furthermore, non-healing ulcers are more likely to form when diabetic neuropathy damages the distal nerves of the feet in addition to micro- and macrovascular damage [2,3,4]. Additionally, changes can be seen in the capillary system of diabetic patients (thickened basement membrane, smaller capillaries, etc.) [5]. Diabetic foot ulcers affect 9.1 to 26.1 million people worldwide yearly [6]. The incidence of diabetic foot ulcers is anticipated to increase as the number of diagnosed DM patients continues to rise annually. Non-healing ulcers can lead to loss of function, mortality, morbidity, excruciating pain, chronic infection, hospitalization, and even surgical intervention, with amputation in some cases [4,5,6]. Diabetes wounds have a significant impact on patients and their families. Diabetic wounds are the most common and debilitating consequences of DM. The high blood sugar of diabetic patients can cause injury to the skin by impairing the capillaries and the peripheral nerves [5,7].

Consequently, many kinds of infection can affect the patient’s body and have a significant impact on patients and their families. As a result, more than 25% of unmanaged or poorly treated cases may necessitate amputation [3]. Because of this, the mortality rate associated with DM is increased [8]. When conditions are physiologically normal, the wounded tissue will begin the process of acute healing on its own [9]. On the other hand, a chronic wound cannot heal if the recovery process is hampered by an underlying pathophysiological condition or pathogenic cause [9,10]. Standard treatment for general wound therapies includes skin perfusion restoration, infection control, and treatment of co-morbidity [9,11]. Although these conventional treatments might accomplish the desired goal of symptom control, there is still a significant gap between the two in terms of the effective treatment of diabetic wounds. In addition, conventional therapies primarily involve the use of dressings; the treatment process is lengthy, and it is simple for diabetic patients to sustain secondary injuries and experience negative psychological and physiological effects. Thus, impaired wound healing in DM has garnered significant interest and has led to the development of numerous approaches, including nanotechnology, primarily to accelerate wound healing [3,9,12,13,14].

*Ocimum sanctum* (family: Labiaceae), known as holy basil, is native to India’s semitropical and tropical regions and now grows natively throughout the eastern tropics of the world [15,16]. Various components of the *Ocimum sanctum* have a long history of use in the traditional medical practices of Ayurveda and Siddha for the treatment of a wide variety of conditions, including infections and dermal disease, as a hepato-protective agent, and as an antidote for snake bites and scorpion stings [15]. In addition, various studies have reported the anti-inflammatory, analgesic, immunostimulatory, and antilipoperoxidant properties of the extract from the leaves of *Ocimum sanctum* [15,16,17,18]. Flavonoids in their leaf extract have an antioxidant activity that plays a significant role in wound healing [19,20]. 

The promising characteristics (physicochemical and biological attributes) of nanomaterials have impacted various fields of health sciences such as regenerative medicine; gene, protein, and drug delivery; diagnosis; bio-imaging, etc. [21,22]. NPs, particularly metallic nanoparticles (MNPs), have recently been the subject of research into their design and development through green synthesis as a potential treatment approach for various pathological conditions, including diabetic wounds [5,21]. Furthermore, green technology-based NPs outperform physical and chemical methods in various ways, including limited use of expensive chemicals, less energy, and the production of eco-friendly products and byproducts [23]. Green synthesis is a bottom-up approach in which costly and toxic chemical agents or solvents are replaced by naturally accessible materials such as whole plants, plant tissue, fruits, plant extract, bacteria, marine algae, and fungi to produce MNPs or metal oxide NPs [24]. Among various MNPs or metal oxide NPs, titanium dioxide (TiO_2_) has been widely adopted as a versatile metallic oxide because of its excellent physical stability and nontoxicity. TiO_2_ has received much attention compared to other antimicrobial agents due to its stability, safety, eco-friendliness, and broad-spectrum antibacterial efficacy [25,26]. In the process of wound healing, titanium dioxide has been shown to have potentially beneficial effects through its antimicrobial and cell growth-stimulating properties [27]. In recent years, TiO_2_ NPs have attracted much interest due to their wide range of applications [26]. TiO_2_ nanoparticles have gained significant attention globally because of their high optical quality, chemical stability, and lack of toxicity [26].

The present study focused on the green synthesis of *Ocimum sanctum* leaf extract-mediated TiO_2_ NPs for managing diabetic wounds. The developed TiO_2_ NPs would be incorporated into a hydrogel-based delivery system as a colloidal dispersion for its topical application in diabetic wounds. Several studies reported that chitosan, a natural biodegradable polymer, has been utilized as an antimicrobial gelling agent for the dispersion of MNPs and other NPs in wound-healing applications [28,29,30]. The mucoadhesive properties of CS would be further helpful to make it stay in contact with the application site for a longer duration and improve the availability of released therapeutics at the wound site [30]. Different researchers have synthesized various MNPs of gold [31], silver [32], nickel [33], and iron oxides [34] utilizing *Ocimum sanctum* leaf extract for different applications. Hence, the preparation of MNPs, particularly TiO_2_ NPs using *Ocimum sanctum* leaf extract through green synthesis technique, and their development into a CS-based topical dosage form is conceptualized as a novel hypothesis that should be validated for its therapeutic efficacy in in vivo diabetic wounds. The current investigation aims to perform the green synthesis of *Ocimum sanctum* leaf extract-mediated TiO_2_ NPs, combine them with the CS gel, and assess their wound healing efficacy in streptozotocin (STZ)-induced diabetic wounds in Wistar rats. 

## 2. Materials and Method

### 2.1. Material

Titanium dioxide was purchased from SRL Chemical (Mumbai, Maharashtra, India). Low molecular weight (average molecular weight of 50,000–190,000 Da), 75–85% deacetylated chitosan (CS) was obtained from Sigma Aldrich, Darmstadt, Germany. Purified water was obtained from Milli-Q^®^ Type 1 Ultrapure Water Systems (Burlington, MA, USA).

### 2.2. Collection and Processing of Plant Sample

The leaves of *Ocimum sanctum* were collected freshly from Dibrugarh, Assam, India. The leaves were washed with clean tap water and then rinsed with double-distilled water to eliminate any dust on the surface [35]. The adequately washed and cleaned leaves were dried at room temperature on clean tissue paper and stored in an airtight container for further processing.

### 2.3. Preparation of Leaf Extract

Twenty-five grams of dried leaves were cut into small pieces and ground in a blender. Next, 20 g of the powdered leaves was blended with 100 mL of purified water in a beaker (250 mL) and boiled for 10 min at 80 °C. After allowing the plant extract to cool to room temperature, it was filtered with Whatman No 1 filter paper [35]. The filtrate was centrifuged at 10,000 rpm for 20 min, and the supernatant was collected and kept at 4 °C for further use.

#### Phytochemical Screening of Active Constituents 

Standard procedures were used to conduct a preliminary phytochemical screening on the aqueous leaf extracts of *Ocimum sanctum* [36]. The presence or absence of phytochemicals (alkaloids, flavonoids, saponins, tannins, terpenoids, steroids, phenols, anthraquinones, protein, and carbohydrates) in the plant leaf extract were tested [37,38,39,40]. 

### 2.4. Green Synthesis of TiO_2_ NPs

We prepared 100 mL of varying concentrations (1–5 mM) of TiO_2_ in round bottom flasks. From this, 80 mL of each concentration of TiO_2_ was placed in 5 different beakers and stirred on a magnetic stirrer at 500 rpm for 2 h at room temperature. Then, 20 mL of plant extract was added separately to each concentration of TiO_2_. This setup was incubated at room temperature for 6 h. Furthermore, TiO_2_ NPs were also prepared by varying the volume of plant extract (5, 10, 15, and 20 mL), keeping TiO_2_ concentration constant. This setup was incubated at room temperature for 6 h. After 6 h, the UV-visible spectrum from each sample was measured, and the optimum concentration of TiO_2_ and the required volume of plant extract was determined. Finally, the biosynthesis of TiO_2_ NPs was monitored using UV-visible spectroscopy [35,41]. TiO_2_ NPs were purified by centrifugation of the solution at 12,000 rpm for 10 min. The supernatant was discarded, and the residue was collected, air-dried, and stored for further studies.

### 2.5. Characterization of TiO_2_ NPs

Biosynthesized TiO_2_ NPs were characterized using Fourier transform infrared (FTIR) spectroscopy, X-ray (XRD) diffraction analysis, scanning electron microscopy (SEM), transmission electron microscopy (TEM), and DLS analysis.

#### 2.5.1. FTIR Analysis

The FTIR spectroscopic method incorporating the principle of attenuated total reflectance (ATR) was performed on FTIR (Alpha II, Brucker, Billerica, MA, USA) spectrometer equipment to characterize aqueous plant extract and green synthesized TiO_2_ NPs. Before the sample analysis, the software’s background (Omnic 8) was scanned, fixing the resolution above 4 and scan time 16. About 1.5 mg of TiO_2_ NPs were placed in the platform, and then the knob of the equipment was placed nicely on the top of the sample. Then, the screw of the knob was turned gently to fixate the sample. Furthermore, the sample scan was carried out in the wavenumber range from 4000–400 cm^−1^ by accumulating 64 scans at 4 cm^−1^ resolution [35].

#### 2.5.2. X-ray Diffraction Analysis

The crystalline nature of TiNPs was confirmed by the XRD pattern obtained from Bruker D8 ADVANCE X-ray diffractometer at 2θ range from 10 to 80°. The sample for XRD measurement was prepared by casting the powder of TiO_2_ NPs on a glass slide and subsequently air-drying it under ambient conditions. The pattern was recorded by CuKα radiation with λ of 1.5406 Å at a voltage of 40 kV and current of 15 mA with a scan rate of 10°/min [42]. 

#### 2.5.3. Surface Morphology Analysis

The surface morphology and size of the green synthesized TiO_2_ NPs were studied using ZEISS SIGMA VP Field Emission Scanning Electron Microscope (FESEM) at 20 kV. For this purpose, the lyophilized sample (Lyophilizer-LABCONCO Make: 710612070) was sonicated in an ultrasonic bath (JSGW, Ambala, India, Model No: 21773-1236/3) for a sufficient amount of time, the smear was made on a glass slide (1 cm × 1 cm) and allowed to dry overnight under vacuum. The slide was then coated with a thin palladium film with the help of argon gas and finally subjected to FESEM. Transmission electron microscopy was performed to precisely determine the size of TiO_2_ NPs. The lyophilized sonicated sample was loaded on a carbon-coated copper grid (400 mesh; Grid hole size 42µm brought from TED PELLA, Inc., Redding, CA, USA) and was allowed to dry overnight in a vacuum and subjected to transmission electron microscopy (TEM-2100 Plus, JEOL) equipped with EDS. Similarly, in the case of TiO_2_ NPs incorporated gel formulation, the smear was made with gel on a glass slide (1 cm × 1 cm) and allowed to dry overnight under a vacuum. The slide was then coated with a thin palladium film with the help of argon gas and finally subjected to FESEM.

#### 2.5.4. DLS Analysis

The particle size distribution, polydispersity index (PDI), and hydrodynamic diameter of biosynthesized TiO_2_ NPs were measured using dynamic light scattering (DLS) techniques with the Nano ZS90 zeta sizer (Malvern Instruments, Malvern, UK) [43,44,45]. A pinch of TiO_2_ NPs was dispersed in 5 mL of double distilled water. From this, 50 µL of the sample was diluted to 5 mL. The diluted sample was subjected to the DLS analysis.

### 2.6. Formulation and Characterization of CS gel Containing TiO_2_ NPs

#### 2.6.1. Preparation of Chitosan Solution

CS solution (2% *w/v*) was prepared by dispersing the appropriate quantity of CS in a 1% *v/v* aqueous acetic acid solution. The mixture was stirred with a magnetic stirrer at room temperature for 24 h. After 24 h, the solution was filtered with Whitman No 1 filter paper [46]. The filtrate was collected and sterilized in an autoclave for 15 min at 121 °C [30]. 

#### 2.6.2. Preparation of CS Gel Containing TiO_2_ NPs

CS gel loaded with TiO_2_ NPs was prepared using the method reported by Afrasiabi et al., with slight modification [30]. Briefly, 1% *w/w* CS gel containing TiO_2_ NPs was prepared by mixing the appropriate quantity of green synthesized TiO_2_ NPs in CS solution (2% *w/v*) with 2 mL of glycerol for 2 h at room temperature using a magnetic stirrer. Finally, pH was adjusted to 6.0 by the slow addition of NaOH (1 M). Similarly, a placebo CS gel was formulated as described above without the TiO_2_ NPs.

#### 2.6.3. Characterization of CS Gel Containing TiO_2_ NPs

The pH, rheology, and spreadability of the CS gel containing TiO_2_ NPs were assessed by the previously reported method [13,43,44]. 

### 2.7. In Vivo Study

Twenty-four albino male Wistar healthy rats weighing 200–300 g were used for this study. The study protocol was approved (approval no. 443-41-53021-DS) by the Institutional Animal Ethical Committee, Najran University, Kingdom of Saudi Arabia. All the animals were kept in clean, germ-free polypropylene cages that met all of the regulations required for a laboratory setting [13,43,44]. They were given access to rodent pellets and water ad libitum.

#### 2.7.1. Diabetes Induction in Rats

STZ was used to induce diabetes in rats. A solution of STZ of appropriate strength was prepared by dissolving a weighed quantity of STZ in freshly prepared ice-cold citrate buffer and injected intraperitoneally at a dose of 55 mg/kg of the body weight [3]. After one week, the rats with blood glucose levels of more than 250 mg/dL were considered diabetic and selected for the in vivo study. 

#### 2.7.2. Creation of Excision Wound in Diabetic Rats

The diabetic rats were anesthetized by intraperitoneal injection of Ketamine HCl (50 mg/kg), and the dorsal side was shaved [12,13]. Using Dettol for cleaning, a circular excision wound of approximately 2.8 cm^2^ in area and 2 mm depth was created using a sterile biopsy punch (Acu punch, Acuderma Inc., Louderale, FL, USA) of 6 mm diameter. All animals were placed into four groups, each having four animals. Group I was the negative control with no treatment, group II was treated with 2% *w/v* CS gel (quantity of the CS gel was based on the body weight of rat), group III was treated with a marketed formulation of 1% *w/w* silver sulfadiazine cream (100 mg on each wound), and group IV was treated with CS gel containing TiO_2_ NPs (dose of TiO_2_ = 20 mg/kg body weight) [12,13]. All the animals of different treatment groups (group II, III, and IV) received the topical application of the aforementioned therapy twice a day for 21 consecutive days. 

#### 2.7.3. Assessment of Wound Healing Activity

The percentage contraction of the wound area, epithelization time, and wound closure time were used to evaluate the healing area [47]. The percentage of wound contraction and the wound area were measured on days 0, 7, 14, and 21 as follows:(1)$$\mathrm{Percentage}\;\mathrm{wound}\;\mathrm{contraction}=\frac{\left(\mathrm{Wound}\;\mathrm{area}\;\mathrm{on}\;\mathrm{day}\;0-\mathrm{Wound}\;\mathrm{area}\;\mathrm{on}\;a\;\mathrm{particular}\;\mathrm{day}\right)}{\mathrm{Wound}\;\mathrm{area}\;\mathrm{on}\;\mathrm{day}\;0}\times 100\;\;\;\;\;\;\;\;$$

#### 2.7.4. Histopathology

On the 21st day of the experiment, the rats were euthanized, and the healed wound area was sampled for histopathological investigation. All the tissue samples from the wounds were fixed in a solution of formalin (10%) and then underwent the standard histological investigation of tissue. Tissues were then embedded in molten paraffin wax, and a 5 μm-thick section was obtained in each case. The sections were stained with hematoxylin and eosin stains. The prepared tissue slide was observed under a microscope. 

### 2.8. Statistical Analysis

The statistical analysis of the findings was carried out with the help of the SPSS software (version 26 SPSS Inc., Chicago, IL, USA). The obtained data were analyzed utilizing one-way ANOVA, followed by Tukey’s multiple comparisons tests. The *p* < 0.05 was considered statistically significant [12].

## 3. Results and Discussion

### 3.1. Phytochemical Analysis

Qualitative phytochemical screening analysis was performed on aqueous leaf extracts of *Ocimum sanctum* to determine the presence of some phytochemicals in the leaves of this plant. The confirmatory test, which involved color changes, precipitate formation, and other confirmations, revealed the presence or absence of phytochemicals such as alkaloids, flavonoids, saponins, tannins, terpenoids, steroids, phenols, anthraquinones, protein, and carbohydrates in the plant leaf extract; these are presented in Appendix A. The results revealed that bioactive compounds such as alkaloids, flavonoids, saponins, tannins, terpenoids, steroids, phenols, protein, and carbohydrates are present in the leaf of *Ocimum sanctum*, while anthraquinone was absent in this plant leaf. These bioactive components may play an important role in the reduction, capping, and stabilization of TiO_2_ NPs [24,26,35,48,49].

### 3.2. Green Synthesis of TiO_2_ NPs

The present investigation has described the *Ocimum sanctum* leaf-extract-mediated preparation of TiO_2_ NPs and their conversion into CS gel through uniform dispersion. TiO_2_ NPs were fabricated at different concentrations (1 mM–5 mM) of the bulk compound using 20 mL of plant extract and were analyzed by UV-visible spectrophotometry. On increasing the volume of the plant extract, there is an increase in the intensity of absorption (Figure 1a). A parallel change in the absorption intensity was observed when different plant extract volumes were used for the biosynthesis of TiO_2_ NPs, keeping the TiO_2_ concentration constant (Figure 1b). From the above results, 80 mL of 5 mM of bulk TiO_2_ solution and 20 mL of aqueous leaf extract were selected for the biosynthesis of TiO_2_. The developed system was further evaluated for wound-healing activity in diabetic rats. UV-visible spectroscopy confirmed the fabrication of TiO_2_ NPs in the aqueous leaf extract of *Ocimum sanctum*. The absorption spectra of the *Ocimum sanctum* leaf-extract-mediated reduced TiO_2_ NPs were observed in the range of 200–800 nm (Figure 1). The peaks recorded at 235 and 320 nm represent the polyphenols [50] and reduced TiO_2_ NPs, respectively, which are preliminary signs for the reduction of TiO_2_ and the formation of TiO_2_ NPs [35,42]. Similarly, in another investigation, TiO_2_ NPs were synthesized using the extracts of *Pouteria campechiana* [35], *Azadirachta indica* [42], and *Luffa acutangular* [48], which were primarily confirmed by UV-visible spectroscopy. In Appendix A, a few potential bioproduction methods for TiO_2_ NPs using phytochemicals are depicted.

### 3.3. Characterization of TiO_2_ NPs

#### 3.3.1. FTIR Analysis

FTIR analysis was used to investigate the identification of functional groups responsible for the capping and efficient stabilization of synthesized TiO_2_ NPs by the aqueous leaf extract of the *Ocimum sanctum*. Prominent peaks were observed at 3396, 2965, 1704, 1626, 1540, 1377, 1047, and 682 cm^−1^ (Figure 2). The observed peaks corresponded to the presence of phenol, flavonoids, carbonyl, amide, aliphatic nitro, aromatic amide, secondary alcohol, and alkyl halide that may have taken part in the green synthesis of TiO_2_ NPs. The presence of these organic groups indicates that the organic source was responsible for reducing TiO_2_ [41,42,48,51]. The peak at 590 cm^−1^ in the FTIR spectra of TiO_2_ NPs is due to the Ti–O–O bond [52]. This investigation supported the capping behavior of the aqueous leaf extract of *Ocimum sanctum* for the produced TiO_2_ NPs, which stabilized the them [41,42,48,53].

#### 3.3.2. X-ray Diffraction (XRD) Analysis

The XRD crystallography pattern of TiO_2_ NPs prepared by green synthesis is shown in Figure 3. The spectra demonstrated the development of anatase-titania with diffraction peaks (2θ) at 27.5° (110), 32.3° (101), 39.9° (011), 48.3° (111), 54.7° (121), 68.9° (031), 74.3° (022), and 76.1° (112). Our results are in agreement with International Centre for Diffraction Data (ICDD) No-21-1272 and 21-1276 [42,49,54]. Furthermore, our results coincide with the different literature reports on the green synthesis of TiO_2_ NPs using a different type of extract [42,48,49,54]. The diffraction pattern and presence of a strong peak indicated the crystallinity of TiO_2_ NPs in anatase form [49,55].

#### 3.3.3. Surface Morphology and EDX (Energy Dispersive X-ray) Analysis

In the current investigation, the surface morphology and size of the green synthesized TiO_2_ NPs were studied using SEM and TEM analysis. Figure 4a,b represent the SEM and TEM images of TiO_2_ NPs prepared by green synthesis. Microscopic investigation revealed that the shape of the developed NPs system is spherical and polygonal. The particle size of the developed NPs was found in the range of 75–123 nm. Similarly, in another investigation, TiO_2_ NPs were synthesized using the leaf extract of Pouteria campechiana with a particle size in the range of 73–140 nm [35]. Figure 4c represents the SEM analysis of the CS gel containing TiO_2_ NPs. It can be observed that TiO_2_ NPs were homogeneously dispersed within the CS gel. The EDX analysis of the developed NPs system showed the presence of titanium and oxygen (Figure 4d). The EDX profile demonstrated a strong TiO_2_ signal and some other weak signals, which might be due to the phytoconstituents from *Ocimum sanctum* leaf extract attached to the surface of TiO_2_ NPs prepared by green synthesis. In this investigation, TiO_2_ NPs exhibited an intense absorption spectrum at 4–5 KeV. Similarly, in another investigation, Srinivasan et al. reported the absorption spectrum of TiO_2_ NPs at 4–5 KeV using *Sesbania grandiflora* leaf extract in EDS analysis [54].

#### 3.3.4. DLS Analysis

The particle size distribution and hydrodynamic diameter of the TiO_2_ NPs were precisely determined using a zeta sizer. The mean particle size and PDI of the TiO_2_ NPs were 130.3 nm and 0.237, respectively (Figure 5a). Furthermore, our investigation observed a significant negative zeta potential (−11.5 mV) for the TiO_2_ NPs prepared by green synthesis utilizing *Ocimum sanctum* leaf extract (Figure 5b). This negative charge on the developed NPs system may be due to the various capping agent from plant sources (*Ocimum sanctum* leaf) present on the surface of TiO_2_ NPs. The existence of a significant amount of negative charge on the surface of developed TiO_2_ NPs, as demonstrated by the high absolute value of zeta potential, would be helpful in the stability of NPs system [14]. Similarly, Lakkim et al. reported that stability of green synthesized NP with zeta potential −15.2 mV [56].

### 3.4. Preparation and Characterization of CS Gel Containing TiO_2_ NPs 

Our investigation used CS to formulate a gel matrix and uniform dispersion of TiO_2_ NPs into it. The CS gel is widely reported for its biocompatibility and desirable safety attributes for living tissues [29,46]. Furthermore, various studies have reported that CS is an excellent antimicrobial agent that can inhibit the growth of different microbes, including bacteria and fungi [29,57,58]. Being hydrophilic, CS can retain the water within its structure and form a gel at acidic pH. Due to its desired consistency and spreadability (Table 1), CS gel was prepared at a concentration of 2% *w/v* [29]. The pH of the formulation was found to be 6.03 ± 0.15, which indicates that it will be safe for use on human skin [12,13,29,43,44]. The spreadability of the CS gel containing TiO_2_ NPs and CS gel was found to be 43.77 ± 1.78 cm^2^ and 46.14 ± 0.69 cm^2^, respectively. The viscosity was observed to be 593.33 ± 6.80 cps and 581 ± 7.50 cps for CS gel containing TiO_2_ NPs and CS gel, respectively. The viscosity values are within desirable ranges, making CS gel suitable for topical use in the wounded area [29].

A comparative rheological study of CS gel containing TiO_2_ NPs compared to CS gel demonstrated similar rheological attributes. Incorporating TiO_2_ NPs into the CS gel matrix did not affect its rheological behavior. CS gel containing TiO_2_ NPs showed pseudoplastic properties with thixotropic behavior (Figure 6). This rheological property for the pharmaceutical dosage forms is desirable for its topical application [12,13,43,44].

### 3.5. In Vivo Study

#### 3.5.1. Wound Healing Activity

As reported earlier, a high dose of STZ results in increased mortality in rats [3]. Therefore, the dose of 55 mg/kg of the body weight was selected to induce diabetes in the experimental animals. Diabetes developed within seven days, and the mortality rate was significantly low (2 deaths/24 rats). However, in an earlier study, diabetes was developed within three days at the same dose, and the reported mortality rate was 3 deaths per 25 rats [3]. The random blood sugar (RBS) measurement before and after the induction of diabetes was found to be 109.81 ± 5.65 mg/dL and 309.56 ± 25.12 mg/dL, respectively. 

TiO_2_ NPs possess a significant therapeutic potential for use in wound healing activity [27]. In our study, TiO_2_ NPs were green synthesized utilizing *Ocimum sanctum* leaf extract and uniformly dispersed into 2% CS gel to assess the improvement in its diabetic wound healing efficacy. Figure 7 represents the in vivo wound healing effect of topically applied CS containing TiO_2_ NPs gel in a diabetic wound model compared to the animals with a diabetic wound treated with a silver sulfadiazine cream (1% *w/w*) available on the market, CS gel (2%), and a negative control without any treatment. The wound contraction area was monitored at pre-specified time intervals. The animals treated with silver sulfadiazine cream (1%) and CS gel containing TiO_2_ NPs displayed no sign of pus development, bleeding, or any evidence of infection during the study period, while animals from group I displayed signs of inflammation (I). In addition, in group I, a firm thrombus swelling and serosanguineous discharge were observed around the wound area. As for the other groups, a rather soft thrombus, reduced edema, and no exudates were observed in group IV, III, and II. On day four, granulation tissue was observed in the animals in group III and IV. However, it was observed late in the animals of group II (11th day of post-wounding). Wounds treated with CS gel containing TiO_2_ NPs and silver sulfadiazine cream (1%) revealed a significant (*p* < 0.05) wound contraction on the seventh day onward, and in the following days, wound contraction was much faster than group II and group I. Complete epithelization time for the animal under treatment with silver sulfadiazine cream (1%) and TiO_2_ NPs containing CS gel were 11.25 ± 0.5 and 10.25 days, respectively. In the animals of group II (2% CS gel), the epithelization time was 14.75 ± 0.5 days, and for the animals in the untreated group (group I), it was 19.5 ± 0.57 days. The complete epithelization time for the diabetic wound in this study was significantly (*p* < 0.05) lower in group III and group IV compared to group I and group II animals. 

Group III and group IV exhibited significantly (*p* < 0.05) enhanced wound closure on the seventh day onward compared to group I and group II. Overall, the wound contraction area was increased in all treatment groups (group II, III, and IV) (Figure 8). 

Compared to group I and group II, a significantly higher (*p* < 0.05) wound closure was observed on the 14th day in group III and group IV. Complete wound healing in group III and group IV was almost accomplished by day 21. The results of wound area (cm^2^) values in the different treatment groups of animals are shown in Table 2.

#### 3.5.2. Histopathology

The histopathological finding of the treated animal after the 21st day of post-wounding is shown in Figure 9. A high incidence of inflammation (I) was observed in the dermal section of group I animals and group III animals. In contrast, low inflammation was observed in group II (2% *w/v* CS gel) and group IV (TiO_2_ NPs containing CS gel) animals, indicating the anti-inflammatory role of chitosan [59]. Hair follicles (H) and keratinization (K) were observed in group II, III, and IV animals. However, it was more prominent in group III and group IV animals, signifying the wound-healing role of silver sulfadiazine cream and the developed formulation system. Granulation tissue (G) was observed in all groups, while untreated animals (group I) showed some blank regions. Blood vessels were also observed in group II and group IV animals, suggesting that CS gel plays its role in angiogenesis and synergizes effect on the wound healing activity of CS gel containing TiO_2_ NPs.

## 4. Conclusions

This study investigates the *Ocimum sanctum* leaf-extract-mediated green synthesis of TiO_2_ NPs and their further incorporation into 2% CS gel to develop TiO_2_ NPs CS gel and explore its wound healing activity in diabetic rats. The *Ocimum sanctum* leaves extract was used as a significant reducing agent for the bioproduction of TiO_2_ NPs. This eco-friendly green synthesis of TiO_2_ NPs would be a godsend for a clean, non-toxic environment. The fabricated TiO_2_ NPs were characterized by UV-visible spectroscopy, FTIR, XRD, SEM-EDX, and TEM analysis. The FTIR analysis revealed the role of aqueous leaf extract *Ocimum sanctum* in the fabrication of NPs. The XRD study showed that the crystallinity of the TiO_2_ NPs is in anatase form. The microscopic investigation revealed the shape of the NPs as spherical, polygonal, and square. The size of the NPs was found in the range of 75–123 nm.

Furthermore, the DLS investigation demonstrated that the mean particle size and PDI of the green synthesized TiO_2_ NPs were 130.3 nm and 0.237, respectively. The green synthesized TiO_2_ NPs were successfully incorporated into a 2% *w/v* CS solution to prepare TiO_2_ NPs CS gel. The developed TiO_2_ NPs CS gel exhibited significant wound healing activity in diabetic rats, confirmed by measuring the wound contraction area and via histopathological investigations of the healed wounds. In light of this, the current investigation ushers in a new era in the drug delivery field by laying the groundwork for exploring the potential of green synthesized TiO_2_ NPs and their incorporation into CS gel for wound healing. 

## Figures and Tables

**Figure 1 molecules-27-07712-f001:**
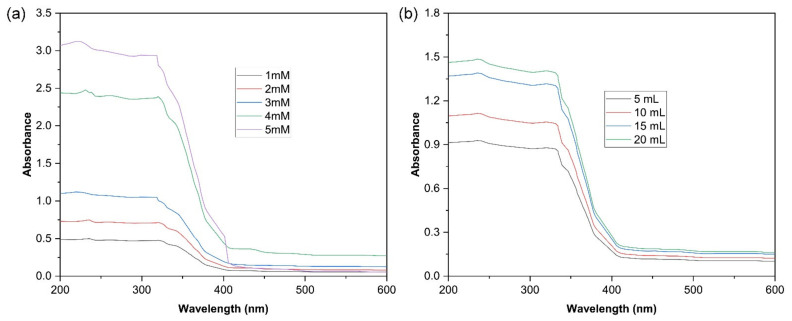
UV-vis spectroscopy revealing the spectrum of TiO_2_ NPs prepared by green synthesis using *Ocimum sanctum* leaf extract (**a**) bulk TiO_2_ (1–5 mM) and using (**b**) aqueous leaf extract (5 mL–20 mL).

**Figure 2 molecules-27-07712-f002:**
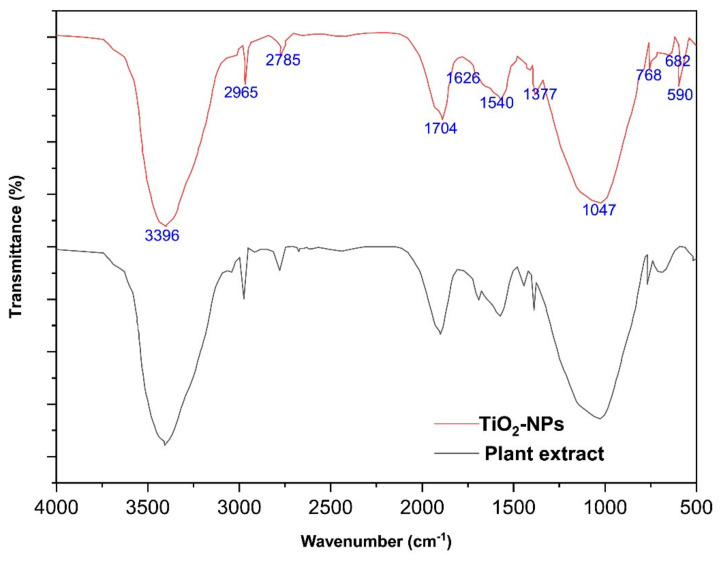
FTIR spectra of TiO_2_ NPs prepared by green synthesis.

**Figure 3 molecules-27-07712-f003:**
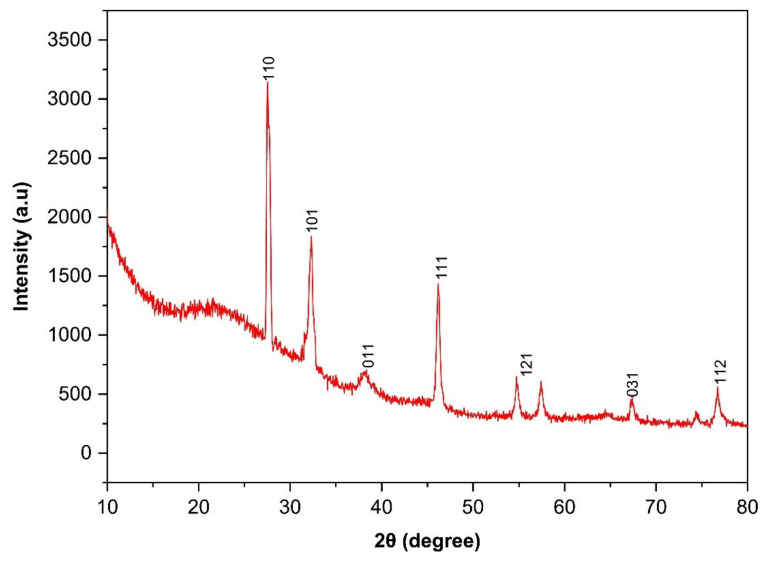
XRD spectra of TiO_2_ NPs prepared by green synthesis.

**Figure 4 molecules-27-07712-f004:**
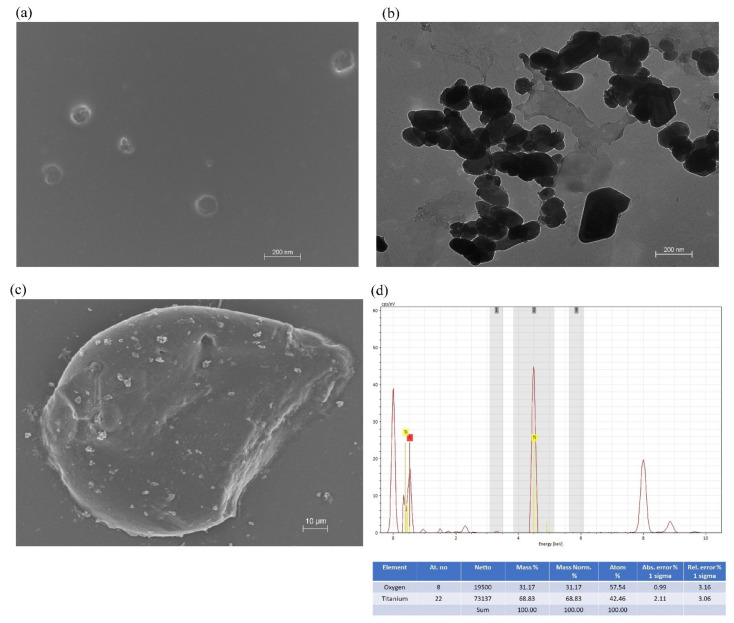
Surface morphology and energy-dispersive X-ray (EDX) analysis of TiO_2_ NPs prepared by green synthesis. (**a**) SEM image of TiO_2_ NPs. (**b**) TEM image of TiO_2_ NPs. (**c**) SEM image of CS gel containing TiO_2_ NPs. (**d**) EDX analysis of TiO_2_ NPs.

**Figure 5 molecules-27-07712-f005:**
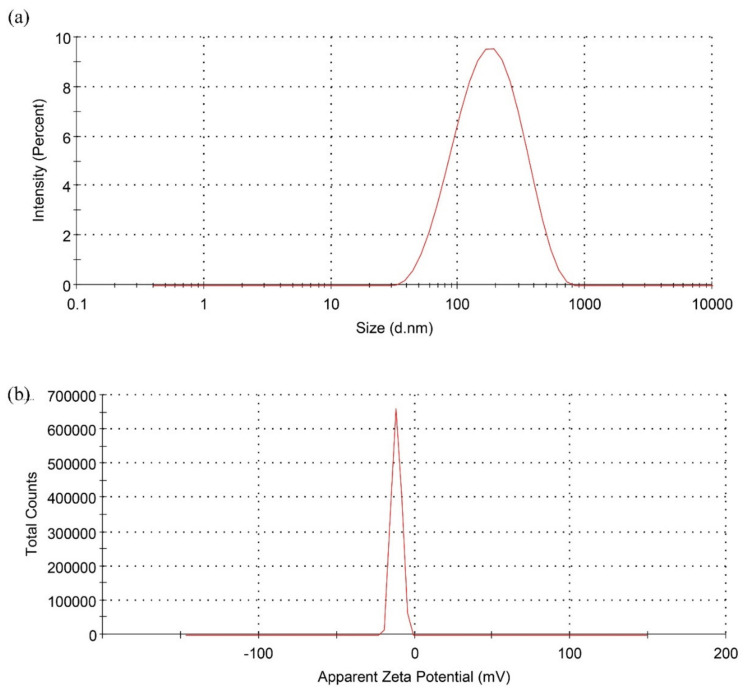
Characterization of TiO_2_ NPs by photon correlation spectroscopy. (**a**) Particle size distribution. (**b**) Apparent zeta potential.

**Figure 6 molecules-27-07712-f006:**
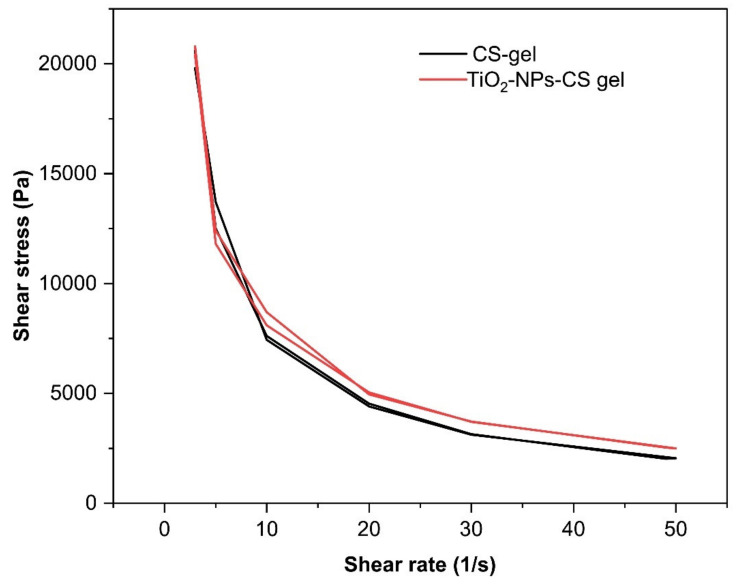
The rheological profile of CS gel and CS gel containing TiO_2_ NPs exhibited pseudoplastic behavior.

**Figure 7 molecules-27-07712-f007:**
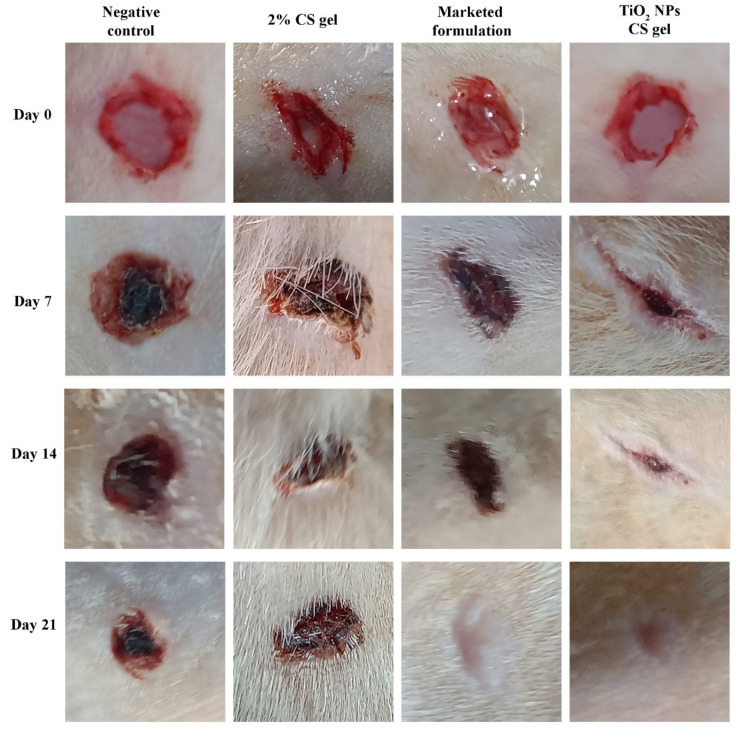
Wound healing efficacy of CS gel, CS gel containing TiO_2_ NPs, and marketed product in diabetic wounds of Wistar rats.

**Figure 8 molecules-27-07712-f008:**
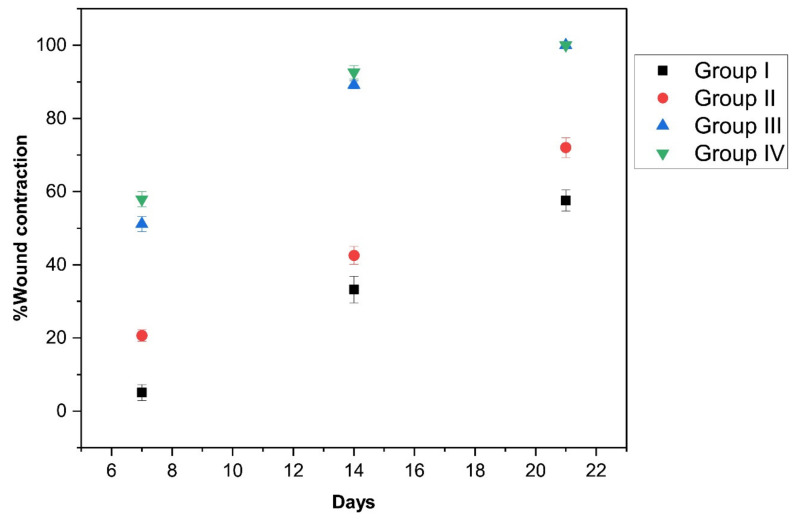
Percentage of contracted wound area in different treatment groups (group I, II, III, and IV) of animals.

**Figure 9 molecules-27-07712-f009:**
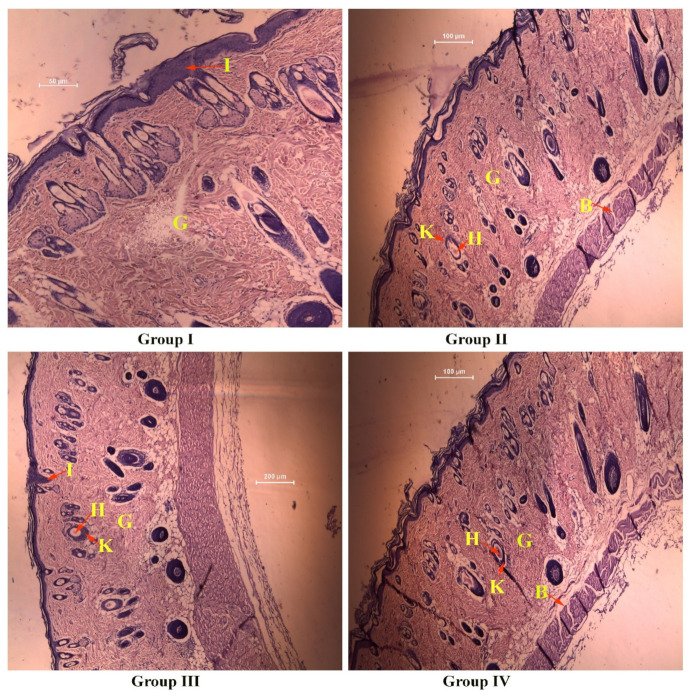
Histopathological changes in diabetic wounds after 21 days of treatments (animals of groups I, II, III, and IV).

**Table 1 molecules-27-07712-t001:** Physical characterization of CS gel and CS gel containing TiO_2_ NPs.

Formulation	pH	Spreadability (cm^2^)	Viscosity (cps)
TiO_2_ NPs CS gel	6.03 ± 0.15	43.77 ± 1.78	595.33 ± 6.80
CS gel (placebo)	5.96 ± 0.20	46.14 ± 0.69	580.97 ± 7.50

**Table 2 molecules-27-07712-t002:** Wound area (cm^2^) in the different treatment groups of animals.

Treatment	Day 0	Day 7	Day 14	Day 21
Group I	28.54 ± 0.17	27.09 ± 0.45	19.04 ± 0.98	12.10 ± 0.78
Group II	28.30 ± 0.05	22.47 ± 0.48	16.25 ± 0.70	7.91 ± 0.50
Group III	28.35 ± 0.13	13.85 ± 0.53	3.06 ± 0.29	completely healed
Group IV	28.35 ± 0.10	11.94 ± 0.49	2.09 ± 0.50	completely healed

## Data Availability

Not Applicable.

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
