# Peer review of "Green Synthesis of Titanium Dioxide Nanoparticles Using Ocimum sanctum Leaf Extract: In Vitro Characterization and Its Healing Efficacy in Diabetic Wounds"

_molecules, 2022, doi:10.3390/molecules27227712_

Round 1

Reviewer 1 Report

The reported work entitled "Green synthesis of titanium dioxide nanoparticles using Ocimum sanctum leaf extract: in vitro characterization and its healing efficacy in diabetic wounds” is interesting. However, the manuscript can be accepted in Molecules after taking my concerns into account, as follows.

1. Many grammatical and typographical errors must be carefully corrected.

2. There are more studies about including metallic nanoparticles in CS as gel formulation. What is new in this study? Please discuss the novelty of your study in the introduction clearly.

2. In the abstract and within all manuscript, the authors must use the term FTIR analysis instead FTIR and XRD instead X-RD.

3. In the section describing Characterization of TiO2 NPs, the authors must correct the first sentence by replacing the apparatus with the method. A detailed description of the analysis methods must be done.

4. Which is the final volume of the CS gel containing TiO2 Nps after correcting the pH value? I am not convinced about the homogeneous dispersion of the nanoparticles within the CS gel. Could the authors provide an evidence of it? I suggest a lyophilization step followed by SEM analysis.

5. A more detailed attribution of FT-IR peaks must be done. In the FTIR spectra, I didn't find any vibration for Ti that should have appeared around 590 cm-1. How can authors justify this statement?

6. What about the size distribution of TiO2 NPs by TEM? From Figure 4, it seems to be a difference between the size and shape of TiO2 NPs determined by the two methods.

7. Which is the pH of CS-gel (placebo)?

8. In the section 2.7.2., why the authors did not express the CS gel containing TiO2NPs in a similar way as did for other formulations? They don’t used the same quantity of CS gel on each wound?  

9. What about the toxicity of the TiO2 NPs and CS-gel formulation? There is a severe concern of human health of manufactured metallic nanoparticles. Some studies have demonstrated that the cytotoxicity of TiO2 nanoparticles was as considerable as the other nanoparticles like nanosilver and nano silicons. I suggest to strength the present study with the cytotoxic assay on TiO2 Nps and CS-gel formulation used for diabetic wound treatment.

Author Response

Authors would like to express their deepest gratitude to the reviewers for critically reviewing the submitted manuscript. It will certainly improve the quality of manuscript. Kindly find here the attached file of Author response

Reviewer 2 Report

General comments & suggestions:

Overall, this manuscript has merit and demonstrates a novel process of creating TiO2 NPs for enhanced wound healing in diabetic patients. My main concern with this manuscript is that the mechanism behind the use of Ocium sanctum leaf extract to synthesize TiO2 NPs is not explained. No characterization of the extract to identify chemical structures of important molecules and active ingredients present, is shown. There is information about the method/process, but the chemistry is not clearly articulated in the manuscript (e.g., when the extract is added to the solution what components of the extract react with the TiO2 solution to reduce it and form NPs, what concentrations of these chemicals are approximately needed for the reaction to complete, what is the yield of the reaction as described here and why the concentrations/ratios used are the optimal)? Because this is a novel approach, I think that a schematic representation of the synthesis showing the steps and the key chemical structures followed my a short discussion would greatly help the reader understand the process, as well as what are the impurities and byproducts of the reaction.

- Are these NPs intended for topical application only? Healing of external or internal wounds, or possibly both and if not why? What about bioaccumulation in the case of larger wounds, or multiple application scenarios on a large number of wounds? Was a toxicity test performed to identify if at high concentrations the material becomes toxic, because toxicity is usually a function of concentration for certain substances and materials.

- Please be consistent with the presentation of the figures/graphs in the entire manuscript. For example, figures 1, 4c and 5 have a different format than figures 2 or 3 (gray background or grits present, no double axes, etc.), the fonts used for figures 1 and 2 look the same, but in figure 3 they appear to be larger, the symbol used for (a.u.) in the y-axes of figures 1 and 3 are not consistent, etc. If a cytotoxicity study is not necessary, then please discuss why in the manuscript. 

Introduction

- Paragraph 2, 3 lines before the end of paragraph, please consider changing “a great deal of interest” to “significant interest”.

- Paragraph 3, line 2, please use subscript for the number 2 and replace “. TiO2 has…” with “. TiO2 has…”.

- Paragraph 3, 3 lines before the end of paragraph, please consider rephrasing “have skyrocketed widespread attention…” to “have gained great attention…”.

- Paragraph 4, 3 lines before the end of paragraph, please add the word “properties” to the sentence, as such: “, and antilipoperoxidant properties of extract from…”.

- Paragraph 5, fourth line, please delete “(NPs)”, it has been already defined in the abstract and used as an abbreviation in the text, making it redundant in this case.

- Paragraph 6, line 4, please specify what “…enhance the availability of released material.” means. Is this for a different type of chitosan-based drug delivery system described in the literature, and why is this property relevant /important in this study?

Materials and Methods

A general note/recommendation for this section is to add some details regarding the experimental conditions used for each characterization method. As an example, it would be very useful and pertinent information to have the number of scans and the resolution used for the collection of the FTIR spectrum presented. Same for the SEM-EDX analysis and the rest of the techniques described herein.   

- In section 2.3. please correct the degrees Celsius from “80° C” to “80 °C”.

- In section 2.4. the preparation of an 80 mL TiO2 solution at 5 mM is been discussed. What was the solvent (acid?) used to prepare this solution? Where there any surfactants (if aqueous solution) used in this step? Please elaborate a bit more on this process.

- In section 2.6. please add a non-breaking space between numbers and the percent symbol, or units where necessary. For example, “CS solution (2% w/v) was…” should be “CS solutions (2 % w/v) was…”. Please check that thought the entirety of the manuscript.

Results and Discussion

- In the caption for figure 1, please add space between “TiO2” and “NPs”. Same for the caption of figure 2.

- In section 3.2.1. in line three, please change “Prominent peak was observed…” to “Prominent peaks were observed…”. Also, it is not necessary to have an accuracy of two decimal places when reporting the wavenumbers for the location of the absorption peaks (e.g., 3396.71 cm-1 could be simply 3396 cm-1).

- In section 3.2.2. in line three, the word “peak” should be plural “peaks”. Also, please add the degree symbol next to the “68.9 (031)” peak.

- In section 3.2.3. the table reporting the percent elemental composition results from the EDX analysis is very small and has low resolution making it difficult to read. I would recommend creating and transferring the data into a new table. Also, the strong absorption peak present in figure 4c at round ~ 0 keV, is it an artefact due to the material of the EDS window in the detector, or is it actual carbon atoms that were not properly washed off the surface of the NPs after the synthesis protocol? SATW windows are commonly used for EDS instruments, and they can give strong background counts around the carbon peak area due to their composition. Please check if the SEM used is equipped with a SATW window detector.

- In section 3.2.4. the discussion related to the negative charge of the NPs is not very accurate nor clear. First, the surface charge required by NPs to maintain a stable suspension in an aqueous system is usually reported at values equal or above ± 30 mV. Also, please elaborate, what does “various capping agents from plant sources” mean? Have you identified any impurities left on the surface of the NPs after the synthesis (e.g., NMR analysis)? Have any stabilizers been used in the NP dispersion (e.g., surfactants)? Are the NPs described in this section bare TiO2 NPs in water, and how long they remain dispersed before aggregates start forming and precipitant is observed.

- In section 3.4.2. please add scale bars for the histopathological images shown in figure 9.  

Author Response

Authors would like to express their deepest gratitude to the reviewers for critically reviewing the submitted manuscript. Kindly find here the attached word file of author response

Round 2

Reviewer 1 Report

The authors have improved the manuscript by supplementary analysis; however, I have some comments as follows:

1. page 3, line 7 – please verify the sentence meaning. It seems that “wound” must be “would”.

2. page 3, line 10 – what do the authors intend to say by “natural biopolymers in the form of chitosan”?

3. page 4, line 1 – I think that the authors intended to use (1mM – 5mM) TiO2 concentrations.

4. Section 2.5. It must be specified the analysis and not the apparatus, so the authors must write; FT-IR spectroscopy, scanning electron microscopy, transmission electron microscopy...

5. Section 2.5.3. The authors must describe the preparation procedure of TiO2NPs-CS gel sample for SEM analysis and accordingly they should discuss the Figure 4c.

6. Even if the authors replaced the FT-IR spectrum figure, they did not answer at my question regarding the lack of Ti vibration that should have appeared around 590 cm-1. Please comment in the manuscript.

In conclusion, I recommend the manuscript to publication after a minor revision.

Author Response

Authors would like to express their deepest gratitude to the reviewers for critically reviewing the submitted manuscript. It will certainly improve the quality of manuscript

Reviewer 2 Report

I would like to thank the authors for adequately addressing all the comments. 

I believe that the manuscript at its current form is in good shape for publication and that the quality of presentation, as well as the scientific soundness are significantly improved. I would suggest that the authors do a final revision and correct any minor spelling errors (subscripts/superscripts were necessary, space before units, etc.) before the final submission.     

Author Response

(The authors gave the same response as above.)
